# Creating a ‘Molecular Band-Aid’; Blocking an Exposed Protease Target Site in Desmoplakin

**DOI:** 10.3390/jpm11050401

**Published:** 2021-05-12

**Authors:** Catherine A. Hoover, Kendahl L. Ott, Heather R. Manring, Trevor Dew, Maegen A. Borzok, Nathan T. Wright

**Affiliations:** 1Department of Natural Sciences, Mansfield University of Pennsylvania, Mansfield, PA 16933, USA; hooverca17@mansfield.edu; 2Department of Chemistry and Biochemistry, James Madison University, Harrisonburg, VA 22807, USA; ottkl@dukes.jmu.edu; 3Department of Physiology and Cell Biology, Wexner Medical Center, Ohio State University, Columbus, OH 43210, USA; Heather.Manring@osumc.edu (H.R.M.); dew.47@osu.edu (T.D.)

**Keywords:** desmoplakin, molecular dynamics, calpain, arrhythmogenic cardiomyopathy

## Abstract

Desmoplakin (DSP) is a large (~260 kDa) protein found in the desmosome, a subcellular complex that links the cytoskeleton of one cell to its neighbor. A mutation ‘hot-spot’ within the NH_2_-terminal third of the DSP protein (specifically, residues 299–515) is associated with both cardiomyopathies and skin defects. In select DSP variants, disease is linked specifically to the uncovering of a previously-occluded calpain target site (residues 447–451). Here, we partially stabilize these calpain-sensitive DSP clinical variants through the addition of a secondary single point mutation—tyrosine for leucine at amino acid position 518 (L518Y). Molecular dynamic (MD) simulations and enzymatic assays reveal that this stabilizing mutation partially blocks access to the calpain target site, resulting in restored DSP protein levels. This ‘molecular band-aid’ provides a novel way to maintain DSP protein levels, which may lead to new strategies for treating this subset of DSP-related disorders.

## 1. Introduction

Arrhythmogenic cardiomyopathy (ACM/ARVC), one of the cardiomyopathies associated with desmoplakin hotspot variants, is a rare (1:2000–1:5000) inheritable disease [1]. ACM is characterized by fibrofatty myocardial replacement, ventricular arrhythmia, and a decrease in ventricular ejection volume, all of which predisposes affected individuals to sudden cardiac death [1,2,3,4,5]. Raising awareness of ACM has led to an increased impetus to discover its molecular underpinnings; the international task force of ACM states, “the definitive cure of [ACM] will be based on the discovery of the molecular mechanisms that are involved in the etiology and pathogenesis of the disease” [6].

Eighty percent of ACM cases are linked to variants in genes that encode desmosomal proteins [1]. The desmosome is an adhesive multi-protein intercellular junction that links the intermediate filament networks (desmin in cardiomyocytes, and keratin in epithelial cells) of neighboring cells [2,7,8]. In the heart, it has been shown that these connections impart tensile strength to cell–cell junctions and assist with intercellular signaling [3,8,9,10]. These functions promote robust cardiomyocyte function, regulation, and contractile synchronization [7,8,9,11]. Conversely, desmosome destabilization is associated with a weakening of intercellular adhesions, altered electrical conductivity, modified signal transduction, and increased adipocyte and fibroblast infiltration of the heart [2,3,8,9,10,12,13].

Desmoplakin (DSP) is an essential protein within the desmosome [7,14]. With the help of the adaptor proteins plakophilin and plakoglobin, DSP connects the cadherin-like proteins desmoglein and desmocollin, to desmin or keratin [7]. DSP also localizes the gap junction connexin proteins to the desmosome and can influence Ras signaling [13]. Thus, DSP’s molecular interactions underscore its physiological role as a nexus for both mechanical coordination and signal transduction between cells [15].

Around 5% of ACM cases are linked to DSP variants [1,13,16]. Many of these variants result in decreased DSP levels, which subsequently inhibit desmosome formation [11,17]. While the effects of some variants on DSP function are obvious (e.g., those that result in a premature stop codon), other variants are associated with neither a loss of mRNA level, ablation of target protein binding, nor global protein instability [17,18,19]. Instead, a subset of variants, including S299R, S442F, R451G, and S507F, decrease cellular DSP levels through increased protein degradation, triggered by a hypersensitivity to calpain proteolysis [17].

Calpain is a ubiquitously expressed calcium-activated cysteine protease [20]. This protease functions by cleaving specific long-lived proteins to modify targeted cellular activities [21,22]. However, in diseases such as Alzheimer’s disease, diabetes, and some heart diseases there is a significant increase in calpain activity resulting in indiscriminate degradation of calpain targets [23,24]. Calpain targeting is poorly understood. Studies using small peptides have revealed a loose consensus sequence, but in situ targeting appears to rely heavily on secondary structure; any putative calpain target must be experimentally verified [25,26,27].

If *DSP* variants produce otherwise functional proteins that are prematurely degraded by calpain, one obvious solution is to block calpain cleavage using calpain inhibitors [17,28]. This should restore DSP levels, restore the desmosome, and reverse the underlying dysregulation that leads to ACM. However, basal calpain activity is essential in maintaining cardiomyocyte function; among other duties, it regulates various intracellular enzymes and transcription factors that lead to cell maturation and longevity [29,30]. Previous in vitro studies showed that impairment of calpain is a common feature in the development of limb girdle muscular dystrophy type 2A (LGMD2A) [31], while the absence of calpain causes myonuclear apoptosis [30]. As an alternative to inhibiting calpain altogether, Barefield et al. removed the calpain target site from myosin binding protein C and found this modification to be cardioprotective [32]. Thus, while calpain inhibitors are not therapeutically useful, targeted molecular modifications of calpain target sites may be viable.

Here, we explore methods that block calpain targeting of DSP specifically, without disrupting other calpain functions. We describe efforts to conceal the calpain target site on DSP through the introduction of a secondary mutation. The secondary single point mutation—tyrosine for leucine at amino acid position 518 (L518Y), neighbors the previously identified DSP clinical variants (S299R, S442F, R451G, and S507F) in space (but not sequence) and acts as a molecular ‘band-aid’, partially occluding the variant-exposed calpain target site. This proof-of-concept approach serves as a template for future work to rescue mutant desmoplakin through a targeted approach.

## 2. Materials and Methods

*Site-directed mutagenesis.* Individual point mutations were introduced into the NH_2_-terminus of human DSP (amino acids 1-883) as described in [17]. The following primers, forward 5′-CCCTCCTCCGAACCCATACGCCGTGGACCTCTCTTGCAAG-3′ and reverse 5′-CTTCCAAGAGAGGTCCACGGCGTATGGGTTCGGAGGAGGG-3′ were used to introduce the L518Y mutation on the background of wildtype (WT) and each clinical mutation (S299R, S442F, R451G, and S507F) using the QuikChange Site-directed mutagenesis kit (Agilent Technologies) according to the manufacturer’s instruction, and mutations were confirmed with Sanger sequencing.

*Protein purification*. Human wildtype and mutant DSP (amino acids 1-883) were expressed in *Escherichia coli* (BL21[DE3]). Wildtype DSP was induced with IPTG at an OD_600_ of 0.6 and grown for an additional 4 h at 30 °C. Mutant constructs were induced for 10 h at 16 °C. Cells were pelleted, resuspended in phosphate buffered saline (PBS) with the Complete Protease Inhibitor (Roche), sonicated and centrifuged. The supernatant was incubated with glutathione sepharose, and protein was eluted using a 10 mM reduced L-glutathione PBS buffer solution. Elution containing protein was collected, concentrated, and applied to a S200 size chromatography column. Protein purity was determined by SDS-PAGE.

*Circular Dichroism (CD)*. CD spectra were obtained using a 3-mm quartz cuvette in a JASCO J-1500 spectrometer in 5 °C increments from 15 °C to 90 °C. Experiments were completed in triplicate, using 1–3 μM protein in PBS, pH 8.0. Spectra were taken from 200 to 300 nm.

*Calpain assays.* Calpain assays were performed as described in [17] with minor modifications. Purified recombinant DSP protein (5 μg) was diluted in an assay buffer (40 mM Tris HCl, 50 mM NaCl, 2 mM DTT), with 10 mM CaCl_2_ and calpain, for a final DSP concentration of 9.97 μM. Reactions were incubated at 37 °C for 30 min, quenched, and analyzed by SDS-PAGE followed by staining with Sypro Ruby (ThermoFisher) total protein stain according to the manufacturer’s instructions. Quantification of the total amount of protein remaining was calculated via densitometry obtained from ImageJ software (NIH). Values for at least 4 independent replicates were averaged for each variant.

*Statistics.*
Data are represented by the mean and SEM. Multiple comparisons were done with 1-way ANOVA and a Tukey HSD test was used to determine statistical significance. In addition, *p* values of less than 0.05 were considered significant.

*Molecular Dynamics*. Computational modeling was done using the human wildtype DSP residues 178–627 (PDB accession 3R6N) [15]. S299R, S442F, R451G, S507F, L518Y, and double mutant models were generated with the “swap residue” command in YASARA using an AMBER14 forcefield and allowed to equilibrate for 60–200 ns in triplicate in explicit solvent at 37 °C, 150 mM NaCl, as previously described [17]. Simulations were surveyed every 100 ps, and the data were tabulated using YASARA macros. The exposed surface area of residues 447–451 was calculated every 2.5 ns for each mutant using the programs get_area in Pymol and free_sasa (Pymol [33]; free_sasa [34]). Calculations and statistics were performed in R v3.4.3 [35], and data were plotted using ggplot2 [36].

## 3. Results and Discussion

### 3.1. L518Y DSP Point Mutation Proximal to the Calpain Target Site Results in No Overt Structural Changes

DSP is a modular protein with three distinct regions in both structure and function. The NH_2_-terminal third of the protein is composed of 6 α-helical spectrin repeats (SRs) and a Src homology 3 (SH3) domain important in tethering DSP to the desmosomal complex [37,38]. The middle portion of DSP consists of a coiled-coil rod domain necessary for homodimerization, and the COOH-terminus contains three plakin repeats and is responsible for intermediate filament binding [34]. Four ACM-linked variants (S299R, S442F, R451G, and S507F) are located within a mutation ‘hot spot’ in the NH_2_-terminal third of the molecule, within ~15 Å of the SH3/SR4 interface [12,17]. These variants are hypersensitive to calpain-mediated degradation through the exposure of a normally occluded calpain target site (Figure 1 residues 447–451) [17]. Of these variants, two (S299R and R451G) are 0–4 Å from the calpain target site and directly stabilize the area around the site (Figure 1); we refer to these as “proximal”. In contrast S442F and S507F are 10–15 Å away from the calpain target site and will be referred to as “distal”.

It is essential to consider several factors when designing a possible “rescue” mutation for disrupting the calpain targeting of DSP. Calpain targeting efficiency is influenced by amino acid type up to seven residues away from the scissile bond [25]; therefore, a potential “rescue” mutation needs to be sufficiently distant in the primary sequence yet still nearby in the tertiary structure. Additionally, it is required that the “rescue” mutation be able to at least partially physically block the calpain target site, and not destabilize DSP significantly. The only site that satisfies all these factors is L518, located on the entering α-helix of SR5, ~5 Å from the calpain target site (Figure 1) [37]. Modelling showed that swapping Leu for Tyr would be both the least disruptive to the overall structure and most likely to adopt a conformation to block the calpain target site. For this and subsequent experiments, we used the independently folded N-terminal spectrin-repeat region of DSP (residues 1-883). As predicted, this point mutation had no effect on the overall structure or stability of this region (Figure 2).

### 3.2. L518Y DSP Point Mutation Results in Moderate Protection from Degradation by Calpain

To determine whether the introduction of L518Y blocks DSP targeting by calpain effectively, we performed calpain cleavage assays on recombinant DSP (amino acids 1-883) with and without clinical variants (Figure 3). Results were consistent with previous findings [17]; the levels of variant DSP proteins are uniformly decreased when compared to the wildtype DSP protein. Herein, we focus on comparisons to variant protein without L518Y and/or calpain (Figure 3, lane 1 for each panel).

The L518Y mutation partially protects DSP variants in both the presence (a stressed system; Figure 3, lanes 2 and 4 of each panel) and absence (an unstressed system; Figure 3, lanes 1 and 3 of each panel) of calpain. However, these trends are complex. The presence of L518Y does not protect the proximal variants in the absence of calpain; S299R/L518Y and R451G/L518Y DSP variants show an ~40% and 26% reduction in protein levels when compared to protein lacking the “rescue” mutation (Figure 3C,D, lanes 1 and 3; and Table 1). Yet in the presence of calpain, L518Y stabilizes both S299R and R451G protein levels; proteins containing the proximal double mutations (S299R/L518Y and R451G/L518Y) are not hypersensitive to calpain (Figure 3C,D, lanes 3 and 4), while the proximal single mutants (S299R and R451G) are (Figure 3C,D, lanes 1 and 2; and Table 1). Conversely, the rescue mutant protects the distal variants in both unstressed and stressed conditions (Figure 3A,B). Specifically, S442F/L518Y and S507F/L518Y degrade to a lesser extent than protein lacking the “rescue” mutation (S442F and S507F). This holds true in both the absence (23% and 131% increase in stability, respectively) and presence (23% and 130%, respectively) of calpain (Table 1). In comparison, the addition of L518Y in a WT background results in greater protein stability in both the absence and presence of calpain (Appendix A). These data are most similar to the proximal mutants (Figure 3A,B). Together, these calpain assay data show that the addition of L518Y partially reverses variant hypersensitivity to calpain in a variant dependent manner. Additionally, these data, along with docking simulations (Appendix A), reaffirm that the putative calpain target site that we previously identified is, in fact, the main site correlated with the loss of variant DSP levels [17].

### 3.3. Tyrosine at Position 518 Is Insufficient to Robustly and Ubiquitously Protect DSP Clinical Variants from Calpain Targeting Due to Its Size, Position, and Mobility

The MD simulation analysis of each DSP variant reveals that the secondary “rescue” L518Y mutation hides the calpain target site by decreasing its exposed surface area, as predicted. This occlusion is evident in all variants except S299R, which already has a smaller exposed surface area compared to its L518Y double mutant (Figure 4A,A’; Table 1 Column 6). For instance, the distal variant S442F shows varying amounts of the calpain target site exposed, but most commonly ~170–210 Å^2^ of the target site is accessible to the solvent (Figure 4B, dark green trace). In contrast, the S442F/L518Y variant displays both less variation and less exposed surface area of the calpain target site, averaging ~160 Å^2^ (Figure 4B, light green trace). For comparison, the wildtype DSP has a surface area profile similar to S442F/L518Y, with ~140 Å^2^ of the calpain target site exposed to the solvent on average. The R451G and S507F variants display similar trends (Appendix A). The surface area of WT does not appreciably change in the presence of L518Y (~140 Å^2^).

While a decreased target site exposure brought about by L518Y provides a logical explanation for the observed DSP clinical variant rescue, we had originally hypothesized that this protection would be quite robust. To probe the mechanism of why the presence of L518Y offers only relatively weak protection against DSP proteolysis, we examined the orientation of L518Y in MD simulations more closely. First, we noted that one face of tyrosine itself is too small (~60 Å^2^) to completely cover the fully exposed calpain target site (~200–300 Å^2^). Additionally, the simulations reveal that L518Y is positioned over the calpain target site only about 30% of the time (Figure 4C). This percentage is similar in both wildtype and mutant DSP simulations, and roughly corresponds to the expected rotamer probability of tyrosine in this orientation (t80^o^) [39]. In hindsight, this rotamer motion is predictable; tyrosine at position 518 is not corralled into a specific orientation through noncovalent bonds, nor are there nearby hydrophobic moieties for the side chain to pack against.

We next examined why L518Y protects the distal variants from degradation more fully than the proximal variants. The calpain target site resides in a loop between the SH3 domain and the beginning of SR5 and is stabilized on three faces by intramolecular interactions (Figure 5A). The L518Y mutation provides some protection on only one of these faces. When viewed through this lens, it becomes clear why L518Y offers differing levels of protection; L518Y partially blocks the calpain target site from one face but fails to protect against the structural erosion that occurs at the other faces, as a result of either the S229R (on a loop of SR4) or R451G (in the SH3 domain) variants, both of which are directly involved in stabilizing the calpain target site in the native protein.

To explore the proximal vs. distal variant stability effects more thoroughly, we examined all noncovalent interactions stabilizing the calpain target site (residues 446–452) (Figure 5B). The differences in proximal versus distal variants can be most clearly seen in the R451 and S507 interaction; the side chains of R451 and S507 directly bond to and stabilize each other, through noncovalent interactions. While one would therefore expect both the R451G and S507F mutations to disrupt the structure equally, simulations of R451G DSP models contain fewer stabilizing interactions (4, on average) than S507F models (6, on average) (Figure 5C and Appendix A). This trend holds true for the other variants as well; proximal variants disrupt the stability of the calpain target site more than distal variants (Figure 5C and Appendix A).

An in-depth analysis of the calpain target site also revealed that in addition to physically occluding one side of the target site, L518Y also stabilizes the proximal but not the distant mutant structure (Figure 5D and Appendix A). Therefore, it appears that having a more bulky side chain at position 518 partially constrains the motion of these residues, increasing the likelihood they will form stabilizing interactions. Conversely, there are no significant changes in the number of noncovalent bonds in the area surrounding the calpain target site between simulations of the distal mutants (S442F and S507F) and their double mutants (S442F/L518Y and S507F/L518Y). Thus, the proximity of each variant in relation to the calpain target site is critical in determining the molecular effectiveness of L518Y in protecting DSP from targeting by calpain.

## 4. Conclusions

Here we provided evidence for a proof-of-concept that physically blocking a calpain target site can rescue DSP levels. We have created a partial “molecular band-aid” in the form of L518Y, a mutation which can impede calpain targeting of DSP. However, careful analysis shows that this band-aid is only partially functional due to its size, position, and mobility. It follows that to fully protect the calpain target site, it will be necessary to develop large molecules or antibodies that bind to larger patches of the exposed surface of DSP. Keeping both the caveats and the successes identified here in mind should aid in creating more effective future compounds that block DSP degradation.

## Figures and Tables

**Figure 1 jpm-11-00401-f001:**
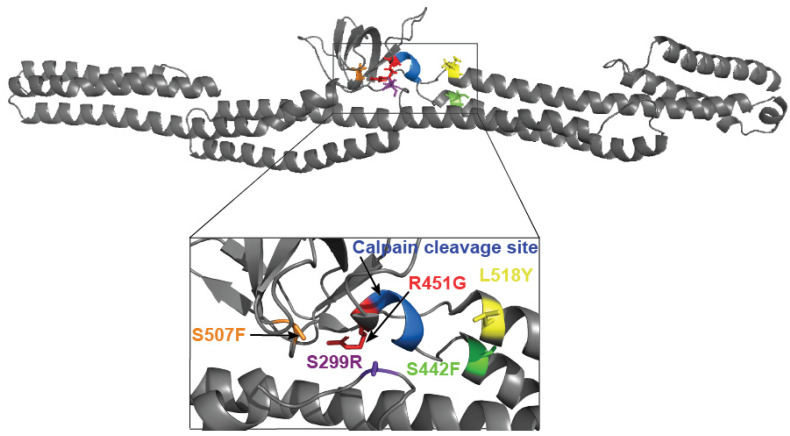
Cartoon representation of Desmoplakin structure. Ribbon model of DSP (178–627; PDB# 3R6N) showing the location of the affected calpain target site (blue) and mutated residues. Red (R451G) and purple (S299R) are proximal variants to the calpain target site, while orange (S507F) and green (S442F) are denoted as distal variants.

**Figure 2 jpm-11-00401-f002:**
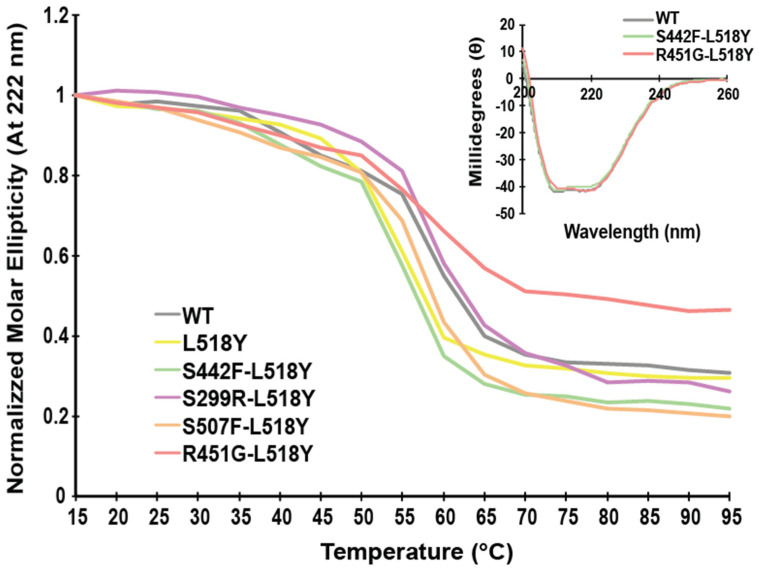
Desmoplakin variants do not influence overall structure or stability. Circular dichroism at 222 nm, normalized to 15 °C for wildtype (WT), and mutant DSP (1-883). All mutants exhibit similar stabilities and near-identical CD spectra.

**Figure 3 jpm-11-00401-f003:**
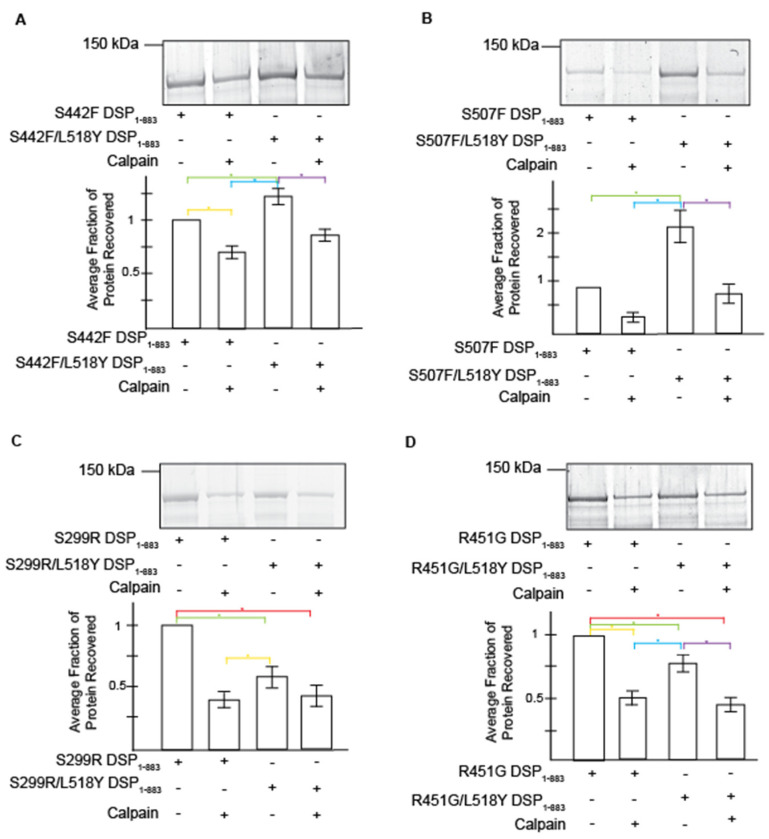
The L518Y mutation partially protects DSP variants from cleavage by calpain. Gels are representative. Graphs show protein recovery ratios as determined by densitometry, for proximal, S442F (**A**) and S507F (**B**), and distal, S299R (**C**) and R451R (**D**) variants. Data are represented by the mean +/- SEM. Statistics were performed with 1-way ANOVA and the Tukey HSD test was used to determine significance. Colored bars represent significance; *p* < 0.05. Statistics were performed and analyzed in sets: S442F Set, *n* = 5; S507F Set, *n* = 6; S299R Set, *n* = 4; and R451G Set, *n* = 7.

**Figure 4 jpm-11-00401-f004:**
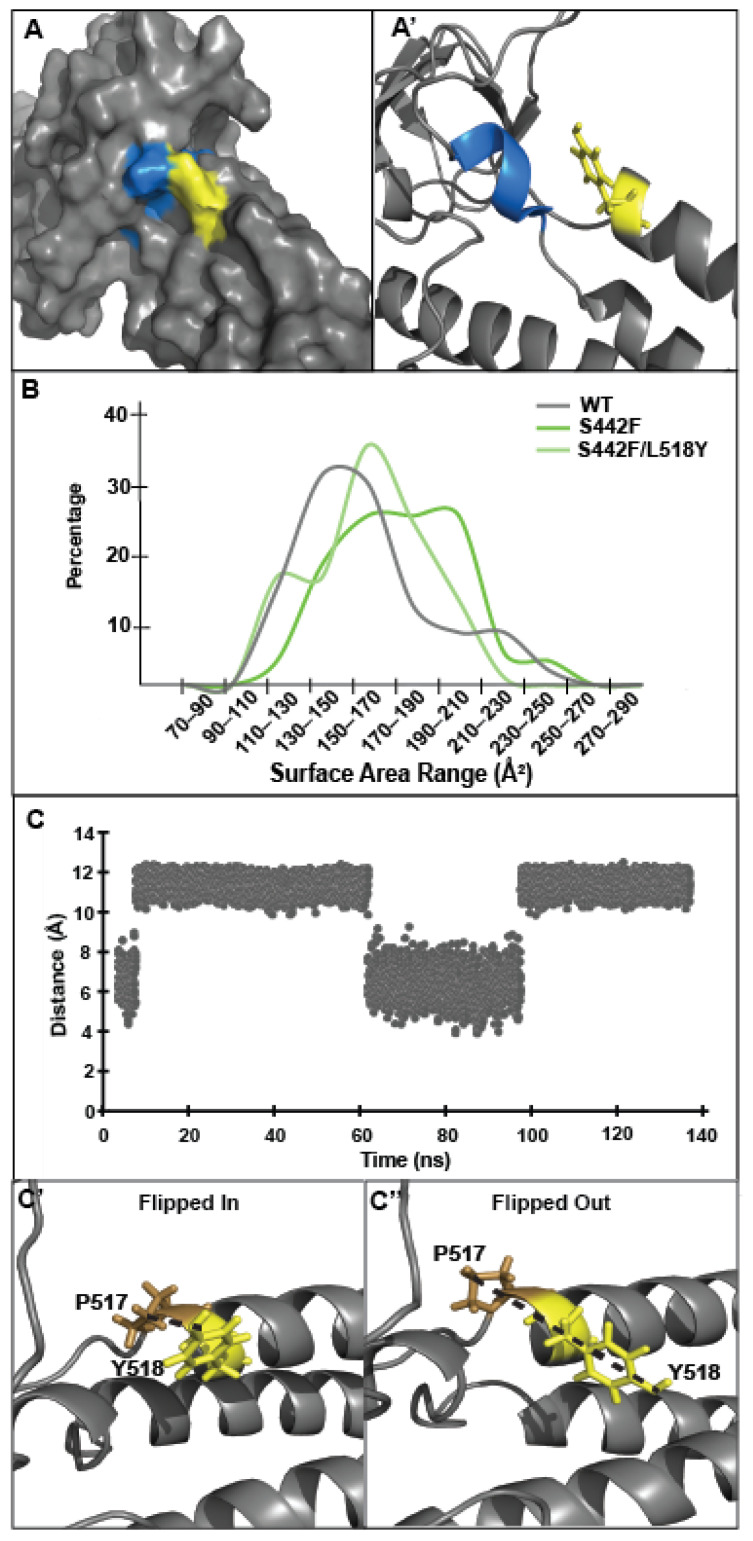
L518Y partially blocks the exposed DSP calpain cleavage site. (**A**,**A’**) L518Y covers part of the calpain target site in the ‘flipped in’ position. (**B**) MD simulations of S442F/L518Y show that the inclusion of L518Y decreases the solvent exposure of the calpain target site. (**C**,**C’**,**C’’**) L518Y in S442F spends roughly 30% of the time in the ‘flipped in’ position, as determined by the distance of L518Y from P517. ‘Flipped in’ is defined as a distance less than 9 Å between the side chains of L518Y and P517.

**Figure 5 jpm-11-00401-f005:**
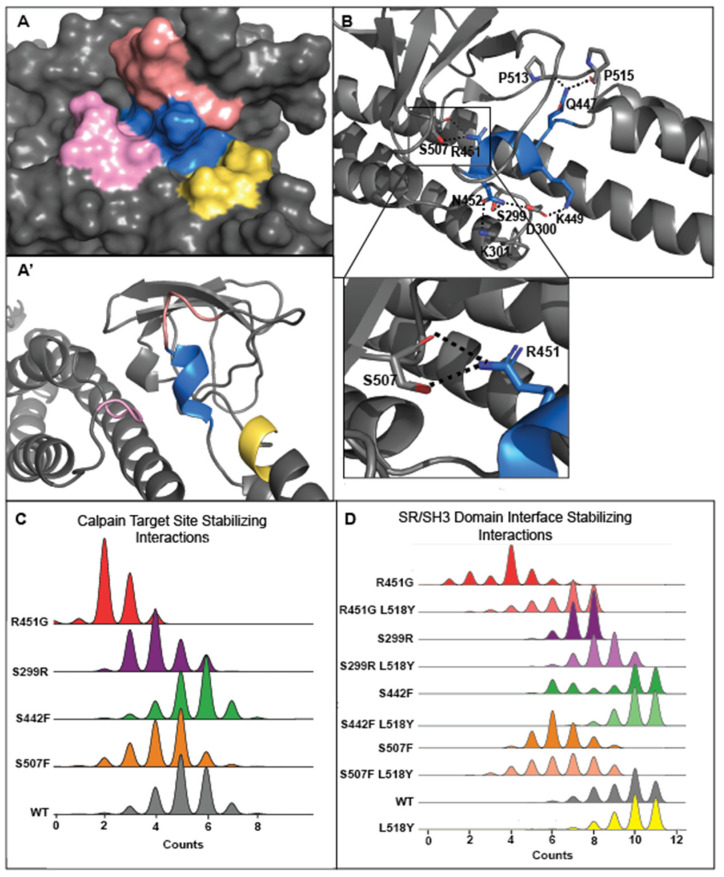
L518Y occludes one of the three faces of the calpain target site. (**A,A’**) The calpain target site (blue) is solvent-accessible on three sides (salmon, pink, and yellow). L518Y is capable of occluding one of these faces (yellow), however the other two faces (salmon and pink) are left exposed. (**B**) The intramolecular interaction between R451G and S507F is integral for the stability of the target site. (**C**) Intramolecular interactions stabilizing the calpain target site decrease in calpain-sensitive variants. (**D**) Intramolecular interactions stabilizing the SH3/SR4 interface, important in occluding the calpain target site, increase in proximal variants in the presence of L518Y. A complete list of these interactions can be found in Appendix A.

**Table 1 jpm-11-00401-t001:** L518Y DSP point mutation results in moderate protection from degradation by calpain and decreased surface area of the calpain target site. Average recovery of DSP variants after incubation at 37 °C for 30 min with and without calpain. Performance of the secondary L518Y mutation was calculated by measuring the percent change compared to its single mutant counterpart. The corresponding solvent-exposed surface area of the calpain target site (447–451) for each variant is also shown. * *p* < 0.05 compared to clinical variant without calpain.

DSP Protein	Avg. Recovery without Calpain	L158Y Performance without Stress	Avg. Protein Recovery with Calpain	L518Y Performance under Stress	Avg. Exposed Surface Area of Target Site (Å^2^)
Proximal Variants					
S299R	1		0.411 ± 0.063 *		148.6 ± 2.9
S299R/L518Y	0.596 ± 0.082 *	↓ 40.1%	0.422 ± 0.085 *	↑ 2.7%	150.6 ± 3.2
R451G	1		0.485 ± 0.054 *		176.4 ± 4.0
R451G/L518Y	0.734 ± 0.064 *	↓ 26.6%	0.482 ± 0.062 *	↓ 0.6%	135.7 ± 4.7
Distal Variants					
S442F	1		0.710 ± 0.060 *		175.8 ± 3.0
S442F/L518Y	1.23 ± 0.07 *	↑ 23%	0.871 ± 0.056	↑ 22.7%	160.7 ± 2.6
S507F	1		0.369 ± 0.064		173.7 ± 3.7
S507F/L518Y	2.31 ± 0.35 *	↑ 131%	0.849 ± 0.171	↑ 130%	163.6 ± 3.0

## Data Availability

MD simulation and CD data can be obtained by contacting NTW (wrightnt@jmu.edu). Original gel images can be obtained by contacting MAB (mborzok@mansfield.edu).

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
