# Peer review of "Creating a ‘Molecular Band-Aid’; Blocking an Exposed Protease Target Site in Desmoplakin"

_jpm, 2021, doi:10.3390/jpm11050401_

Round 1
Reviewer 1 Report
Based on molecular dynamics simulations and a small amount of experimental data, the authors suggest that “physically blocking a calpain target site can rescue DSP levels” and highlight the L518Y mutation as being able to “impede calpain targeting of DSP.” The concept is interesting and important, and the simulation studies are convincing. However, there are major weaknesses in the experimental data that need to be addressed.
- Experimental evidence for the primary claim that the L518Y mutation protects Dsp from calpain-dependent degradation is derived from SDS-PAGE assays (Figure 2). However, important concerns remain to be addressed with regard to these:
- Studies were performed on a peptide corresponding to an N-terminal fragment of Dsp rather than full length protein.
- The methods section states that “at least 3 independent replicates” were performed for each variant. Please indicate how many experiments were performed in each case. Which cases, if any, were statistically significant?
- Control data should be added showing stability of wild-type Dsp.
- Given the importance of this result to the manuscript’s primary claim, orthogonal validation is needed, preferably in a cell or tissue system expressing full-length Dsp protein.
- The simulations assess solvent accessibility of the calpain binding site. Were any docking studies performed between Dsp and calpain?
- What effects, if any, did the secondary mutations have by themselves on calpain binding site accessibility?
Minor Concerns:
- Figure 1: Consider splitting up or enlarging the ribbon model to make it clearer. As it is, it is quite busy.
Author Response
Wethank reviewer 1 for his/her comments and suggestions. We have taken all weaknesses into consideration and have improved the manuscript accordingly. Comment 1: “Experimental evidence for the primary claim that the L518Y mutation protects Dsp from calpain-dependent degradation is derived from SDS-PAGE assays (Figure 2). However, important concerns remain to be addressed with regard to these:Studies were performed on a peptide corresponding to an N-terminal fragment of DSP rather than full length protein.”Response: This is a great suggestion, however, the full-lengthprotein is too large to work with in the recombinant form(~220kDa)and would require a cell or tissue system, which is planned for future studies. The manuscript now explicitly states on page 5, lines172-173: “For this and subsequent experiments, we used the independently-folded NH2-terminal spectrin-repeat region of DSP (residues 1-883).” The construct length is now included in figure legend 2 as well.Comment 1 continued: “The methods section states that “at least 3 independent replicates” were performed for each variant. Please indicate how many experiments were performed in each case. Which cases, if any, were statistically significant?”
Response: We apologize for these oversights and have now included statistical significance and the number of replicates (n) for each set of proteins in Figure 3 and its corresponding legend; page 7, lines 239 and 243-246. Statistical significance was also added to Table 1, page 8, lines 258-259Comment 1 continued: “Control data should be added showing stability of wild-type DSP.”Response: These data are now included as Figure S1 along with the following description that appears in the text on page 7, lines 231-233: “In comparison, the addition of L518Y in a WT background results in greater protein stability, both in the presence and absence of calpain (Figure S1). These data are most similar to the distal mutants(Figure 3 Aand B).”Comment 1 continued: “Given the importance of this result to the manuscript’s primary claim, orthogonal validation is needed, preferably in a cell or tissue system expressing full-length Dsp protein.”Response: While the reviewer poses a legitimate question here, we simply do not have the time nor the resources to currently perform this experiment. We agree that orthogonal techniques are, as a rule, necessary for any kind of scientific experiment; for this reason, we relied on both gel-based assays and computational techniques that both pointed to the same finding-that the L518Y mutation partially rescues mutant DSP from calpain-dependent degradation through an occlusion event. The inclusion of cellular (or additional biochemical) assays are the obvious next steps to both validate and further this line of inquiry. Given the challenging nature of this workand timeline for resubmission, we will include such experiments in future publications. Comment 2: “The simulations assess solvent accessibility of the calpain binding site. Were any docking studies performed between DSP and calpain?”Response: To address this question, we have performed docking experiments. This idea has been incorporated into Figure S2. Here, we show that calpain and desmoplakin can be docked without significant steric hindrance, and that a local unfolding event is necessary for this docking to occur. The text states this on page 7, line 236.Comment 3: “What effects, if any, did the secondary mutations have by themselves on calpain binding site accessibility?”Response: To answer this question, we include the following sentence on page 9, lines 276-277. “The surface area of WT does not appreciably change in the presence of L518Y (~140 Å2).”Comment 4: “Figure 1: Consider splitting up or enlarging the ribbon model to make it clearer. As it is, it is quite busy.”Response: This has been addressed and fixed. Please see new Figures 1 a

Reviewer 2 Report
Arrhythmogenic cardiomyopathy is a kaleidoscopic pathology with different phenotypic manifestations. The complexity and interactions of the desmosome justify the heterogeneity of the phenotype. In the work the interaction with calpain protein was analyzed. Although target blocking for calpain protein is not completely effective in normalizing desmoplakin levels, the experiment provides an encouraging direction for therapies. I suggest accepting the work in present form.
Author Response
Thank you for your kind comments.
Round 2
Reviewer 1 Report
The authors have suitably revised the manuscript based on the initial review.